# Crystal structure prediction by combining graph network and optimization algorithm

Guanjian Cheng[1,2], Xin-Gao Gong[2,3] & Wan-Jian Yin [1,2,4 ✉]

Crystal structure prediction is a long-standing challenge in condensed matter and chemical science. Here we report a machine-learning approach for crystal structure prediction, in which a graph network (GN) is employed to establish a correlation model between the crystal structure and formation enthalpies at the given database, and an optimization algorithm (OA) is used to accelerate the search for crystal structure with lowest formation enthalpy. The framework of the utilized approach (a database + a GN model + an optimization algorithm) is flexible. We implemented two benchmark databases, *i.e.*, the open quantum materials database (OQMD) and Matbench (MatB), and three OAs, *i.e.*, random searching (RAS), particle-swarm optimization (PSO) and Bayesian optimization (BO), that can predict crystal structures at a given number of atoms in a periodic cell. The comparative studies show that the GN model trained on MatB combined with BO, *i.e.*, GN(MatB)-BO, exhibit the best performance for predicting crystal structures of 29 typical compounds with a computational cost three orders of magnitude less than that required for conventional approaches screening structures through density functional theory calculation. The flexible framework in combination with a materials database, a graph network, and an optimization algorithm may open new avenues for data-driven crystal structural predictions.

[1] College of Energy, Soochow Institute for Energy and Materials InnovationS (SIEMIS), and Jiangsu Provincial Key Laboratory for Advanced Carbon Materials and Wearable Energy Technologies, Soochow University, Suzhou 215006, China. [2] Shanghai Qi Zhi Institute, Shanghai 200030, China. [3] Key Laboratory for Computational Physical Sciences (MOE), Institute of Computational Physical Sciences, Fudan University, Shanghai 200438, China. [4] Light Industry Institute of Electrochemical Power Sources, Soochow University, Suzhou 215006, China. ✉email: wjyin@suda.edu.cn

Predicting crystal structure at a given chemical composition *prior* to experimental synthesis has attracted significant interest in condensed matter science. Earlier attempts based on empirical rules provided qualitative descriptions of structures, for example, Pauling's five rules for ionic crystals[1], Goldschmidt's tolerance factor for perovskite formability[2], and dimensional descriptors to classify the zinc-blend (ZB)/wurtzite (WZ) and rock-salt (RS) structures for binary semiconductor compounds[3,4]. Owing to reliable energy calculation via density functional theory (DFT), the current state-of-the-art approaches for crystal structure prediction (CSP) mainly combine DFT calculations with structural searching algorithms such as (quasi-) random search[5,6], simulated annealing[7], genetic algorithm[8-11], particle-swarm optimization (PSO)[12,13], and differential evolutionary process[14]. These approaches extensively explore the structural candidates via searching algorithms and by adopting DFT-calculated energy as a stability metric. The necessary DFT calculations involve the evaluation of numerous structural candidates in the process of structure searching and are thus time-consuming. For example, 70 and 120 DFT structural optimizations are required to determine the ZB structure of GaAs (eight atoms in the cell)[10] and α-quartz structure of $SiO_2$ (six atoms in the cell)[12], respectively.

The advancement of machine learning (ML) in materials science has recently focused on its applications in predicting materials properties such as formation enthalpies ($\Delta H$)[15,16], Gibbs free energies[17], bandgaps[18,19], wave function and electron density[20], X-ray absorption spectra[21], and phase transitions[22]. The accuracy of this approach is close to that of quantum mechanics calculations; however, the computational costs are orders of magnitude lower. In addition to the influence of compositional atoms, the influence of their spatial arrangement, i.e., crystal structure, on materials properties has recently been analyzed via structural characterizing approaches such as the Wyckoff-species matrix-based method[23], Voronoi tessellation method[24], and graph network[18,19]. A crystal (Crys) can be represented by a vector ($\{v_i\}_{i=1,N}$, $\{R_i\}_{i=1,N}$, $L$), where $\{v_i\}$ and $\{R_i\}$ are elemental features and coordinates of the *i*th atom, $N$ is the total number of atoms in a periodic cell, and $L$ is the vector ($a$, $b$, $c$, $\alpha$, $\beta$, $\gamma$) defining the cell shape. In these approaches, crystal structures are transformed to a physically meaningful and algorithm-readable data formats, such as a symmetry-invariant matrix[23], bond configurations[24], or crystal graphs[18], enabling the establishment of a correlation model between a crystal and its formation enthalpy as follows:

$$\Delta H = f(Crys(\{v_i\}_{i=1,N}, \{R_i\}_{i=1,N}, L)) \qquad (1)$$

In principle, CSP can be efficiently performed using Eq. (1) by optimizing ($\{R_i\}_{i=1,N}$, $L$) to minimize $\Delta H$ at a given $\{v_i\}_{i=1,N}$. This approach replaces DFT calculations with the ML model; therefore, it has the potential to significantly accelerate the CSP.

Despite this potential advantage, the practical approach of ML-based CSP still has challenges[25]. First, the ML model should have a sensitive response to the crystal structure; therefore, the fixed-structure model[15,26] and symmetry-invariant model[23], which have a constraint on the crystal structures, are inapplicable or limited in determining the ground state structure (GSS) that may have arbitrary cell shape and atomic coordinates. Second, the high accuracy of DFT calculations benefit from the systematic cancellation of errors relative to the experiment, and the claimed DFT-level accuracies of the ML models are obtained from training data composed of stable crystal structures[27]. The extension of ML models to structural searching is questionable because most structural candidates in the searching process are metastable or unstable, and their relative energies are crucial in

determining the GSS. Finally, an appropriate optimization algorithm compatible with the ML model is required.

In this study, we constructed a framework that establishes a graph network (GN) model between crystal structures and their formation enthalpies at the given database, and this GN model was then combined with an optimization algorithm (OA) for CSP. The framework (a database + a GN model + an OA) is flexible that allows variance in materials database, crystal graph representation, and OA. In this study, we adopted GN developed by Chen et al.[19] as it was designed for both molecules and crystals, facilitating the future extension of the framework to molecules. The Open Quantum Materials Database (OQMD)[28] of version 1.3 and Matbench dataset of formation energy (MatB)[29], have been used separately to train the GN model and random searching (RAS), PSO and Bayesian optimization (BO) has been implemented as OAs. The performance of different combinations have been investigated and compared to predict the crystal structures of 29 octet binary compounds as listed in Table 1, including group IV crystals (C, Si), group I–VII crystal (I = Li, Na, K, Rb, Cs; VII = F, Cl), group II–VI crystal (II = Be, Mg, Ca, Sr, Ba, Zn, Cd; VI = O, S) and typical photovoltaic semiconductors GaAs, CdTe and $CsPbI_3$ (an inorganic representative for perovskite photovoltaics). The comparative studies show that the GN model trained on MatB combined with BO, i.e., GN(MatB)-BO, can predict crystal structures with the best accuracy and extremely low computational cost. The flexibility of graph network, database, and optimization algorithm in the approach facilitate further development and improvement of this approach. This study may open a new avenue for data-driven crystal structural prediction.

## Results

**Crystal graph.** In the original GN[30], a graph is defined by three ingredients, i.e., nodes ($v_i$), edges connecting nodes ($e_k$), and the global attributes ($u$), which are naturally borrowed to crystal graph as atoms, pairs, and macroscopic attributes (e.g., pressure, temperature)[19,31]. Considering that multiple atoms and pairs exist in a crystal, crystal graph is numerically represented by G($\{v_i\}_{i=1:nv}$, $\{e_k\}_{k=1:ne}$, $u$), where $v_i$ and $e_k$ are the elemental and pair attributes of *i*th atom and *k*th pair, and nv and nk are the number of atoms and pairs, respectively, in the cell. In MEGNet[19], $v$ and $e$ are the atomic numbers and spatial distance, represented by $N_v$- and $N_e$-dimensional vectors ($N_v$ and $N_e$ are hyperparameters) learned from model training, respectively. Accordingly, an embedding layer with a $N_v \times nv$ matrix (Fig. 1c) was added after atomic attribute $\{v_i\}$ as input for GN (Fig. 1j). A nv × nv × $N_e$ matrix (Fig. 1d) was added after $\{e_k\}$ (Fig. 1k), where nv × nv represents the pair connectivity between two atoms and each pair is represented by an expanded distance with Gaussian basis numerically represented by $N_e$ points. In comparison to the fixed features, $N_v$- and $N_e$- dimensional vectors can be considered as elemental and pair features that were learned during the model training process. The learned elemental embeddings have been shown to encode the elemental periodicity and can be transferred to predict different properties[19].

**Database and data split.** Two benchmark datasets, OQMD of version 1.3[28], and MatB[29] have been used for GN model training and evaluation. For OQMD, data cleaning was performed to exclude data with incomplete information and restrictions: (i) the number of atoms in the unit cell (<50), (ii) PBE as exchange-correlation functional, and (iii) kinetic energy cutoff (520 eV), making data as reliable and comparable as possible. Accordingly, more than 320,000 data points have been obtained, including ~40,000 experimentally known ones and ~280,000 hypothetical

**Table 1 The performance of GN-OA with different combinations of databases (OQMD and MatB) and optimization algorithms (RAS, PSO, BO) for crystal structure prediction of 29 typical compounds.**

| Compounds | OQMD | | | MatB | | |
|---|---|---|---|---|---|---|
| | RAS | PSO | BO | RAS | PSO | BO |
| LiF | | √ | √ | √ | | |
| NaF | √ | √ | √ | √ | | √ |
| KF | √ | | √ | √ | | √ |
| RbF | | | | √ | | √ |
| CsF | | | | √ | | √ |
| LiCl | | | | | | |
| NaCl | √ | √ | √ | √ | | √ |
| KCl | √ | | | √ | | √ |
| RbCl | √ | | | √ | √ | √ |
| CsCl | √ | | | √ | | √ |
| BeO | | √ | √ | | | √ |
| MgO | √ | | √ | √ | √ | √ |
| CaO | √ | | √ | √ | √ | √ |
| SrO | √ | | √ | √ | √ | √ |
| BaO | √ | | √ | √ | | √ |
| ZnO | | √ | √ | | √ | √ |
| CdO | | | | √ | | √ |
| BeS | √ | √ | √ | √ | √ | |
| MgS | √ | | | | | |
| CaS | √ | | √ | | | √ |
| SrS | √ | | | | | |
| BaS | √ | | √ | √ | √ | √ |
| ZnS | √ | | √ | | | √ |
| CdS | | | √ | | | |
| C | √ | | | √ | | √ |
| Si | | | | | √ | √ |
| GaAs | | | √ | | | √ |
| CdTe | √ | | √ | √ | | √ |
| CsPbI₃ | | | | | | √ |
| Accuracy | 62.1% (18/29) | 20.7% (6/29) | 62.1% (18/29) | 72.4% (21/29) | 27.6% (8/29) | 86.2% (25/29) |

The tick mark means that the GN-OA approach is able to correctly predict the ground-state structures within 5000 iteration steps.

ones, covering 85 elements, 7 lattice systems, and 167 space groups. For MatB, we used the Matbench v0.1 dataset[29] that is derived from data cleaning in Materials Project. For properties of formation energy, it included ~132,000 data points, covering 84 elements, 7 lattice systems, and 227 space groups. The distributions of the number of elements and atoms in each database are shown in Fig. 1a, b. For both OQMD and MatB, the same ratio of data split has been adopted, i.e., training set (50%), validation set (12.5%), and test set (37.5%), to construct GN models for CSP. In all the training, validation, and test process, the data of 29 binary compounds studied in this work, have been excluded.

**GN model**. As shown in Fig. 1h, the GN model was constructed to establish the correlation between the crystals and their formation enthalpies[19]. Crystal graph represented by matrix $\{v_i\}$ (Fig. 1c, j) and $\{e_k\}$ (Fig. 1d, k) are the input of GN model and formation enthalpy $\Delta H$ is the output. There could be $m$ MEGNet layers ($m$ is hyperparameter) that make up the MEGNet blocks (Fig. 1l) to update the matrix $\{v_i\}$ and $\{e_k\}$. The set2set layers (Fig. 1m, n) are used to learn a representation vector from the matrix $\{v_i\}$ and $\{e_k\}$. Then use the concatenate layer (Fig. 1o) to combine these vectors, go through a fully connected layer (Fig. 1p) composed of $l$ dense layers ($l$ is hyperparameter), and get $\Delta H$ (Fig. 1q). Since the symmetries and invariances are included in the current GN model and the pair features are established on the connectivity between two atoms. Cell rotation or symmetry permutation would not change the features, and thus, the GN

model. We train GN model using the data in two respective databases, i.e., OQMD[28], and MatB[29], leading to two different GN models, GN(OQMD) and GN(MatB). By optimizing hyperparameters in Supplementary Table 1, the best performing one in each model was selected to minimize the errors between the GN-predicted and DFT-calculated $\Delta H$ on the test set as the results shown in Fig. 2. The results show that GN(OQMD) has less MAE (16.07 meV/atom) than GN(MatB) (31.66 meV/atom). MAE of GN(MatB) is close to the previous report of 32.7 meV/atom[29]. Such a tiny difference of 1 meV on the same MatB dataset may originate from different data split. The insets in Fig. 2 show a systematic decrease of the MAE as the number of training data. Better performance of OQMD can be ascribed to its larger database (~320,000 DFT-calculated data for inorganic compounds), which is more than twice than MatB. Despite less MAE of GN(OQMD), as shown later, its performance on CSP is inferior to GN(MatB), indicating possible overfitting of GN(OQMD).

**Symmetry constraint**. The wealth of experimental data shows that most of the crystal structures at low temperature have symmetry operations[32] and adding symmetry constraint would accelerate CSP. Meanwhile, most crystal structures in training data, either OQMD or MatB, are symmetrical (with space group spanning from P2 to P230). In this work, we treat CSP with symmetry constraint, by adding two additional structural features, crystal symmetry $S$ and the occupancy of Wyckoff position

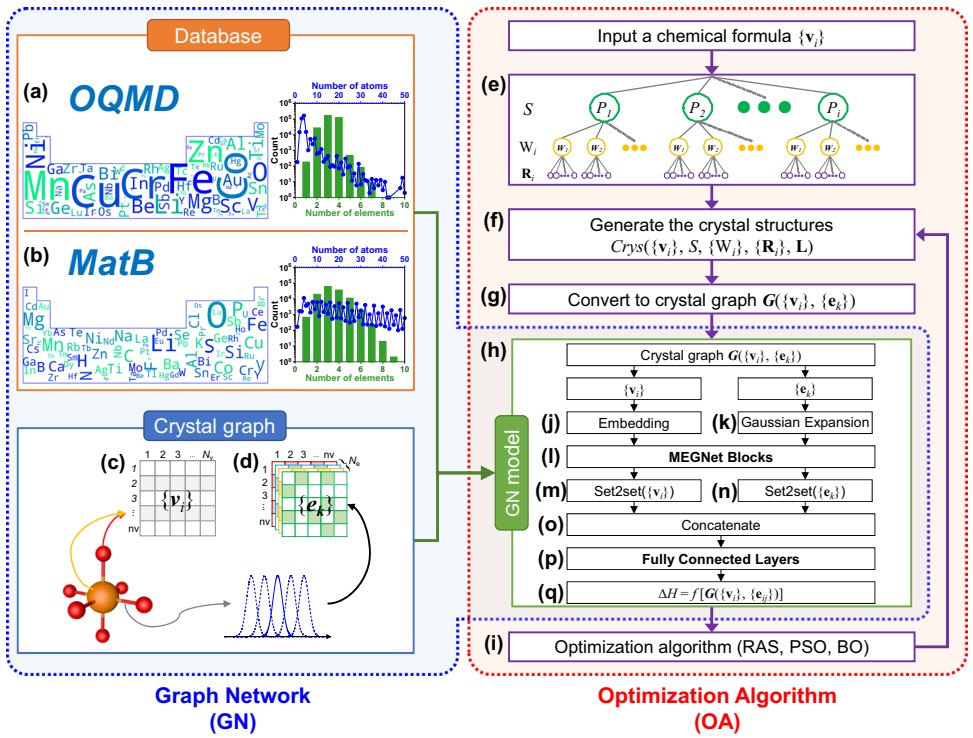

**Fig. 1 Flowchart of GN-OA approach.** Two databases, OQMD and MatB, have been separately used for training GN model and the elemental appearance, the distributions of number of elements and atoms in each database are shown in **a** OQMD and **b** MatB with quantitative number shown in Supplementary Figs. 1 and 2, respectively. For crystal graph, the input atomic feature is **c** embedded atomic number (from 1 to Nv) for each compositional atom (from 1 to nv) and **d** the pair feature is Gaussian-expanded distance (from 1 to Ne) for each pair connecting atom (**i**) (from 1 to nv) and (**j**) (from 1 to nv). The structural generation part includes **e** symmetry constraint, **f** structural generation, **g** crystal graph conversion. The GN model **h** combines **j** embedded atomic number, **k** Gaussian-expanded pair distance, **l** the MEGNet blocks, the set2set layers for **m** atomic number and **n** pair distance, **o** concatenate layer, **p** fully connected layer to obtain **q** the correlation model between a crystal and its formation enthalpy. **i** Optimization algorithm block.

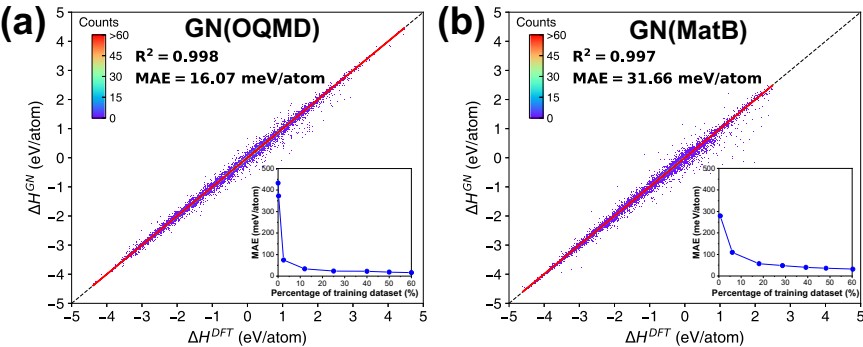

**Fig. 2 The performance of GN model.** GN-predicted and DFT-calculated formation enthalpies in **a** OQMD and **b** MatB. The mean absolute error as a function of percentage of training crystals.

$W_i$ for the i*th* atom, which are chosen through 229 space groups and associated 1506 Wyckoff positions[33]. As shown in Fig. 1e, the procedure firstly chose a symmetry $S$ among P2 to P230 and then the lattice parameters **L** are generated within the chosen symmetry. Secondly, a combination of Wyckoff positions $\{W_i\}$ at given symmetry is selected to meet the number of atoms given in a cell. The atomic coordinates $\{R_i\}$ are then given by the selected Wyckoff position $\{W_i\}$ and lattice parameters **L**. Space group $S$ and corresponding $\{W_i\}$ are variables upon optimizations during CSP with symmetry constraint to generate Crys($\{v_i\}$, $S$, $\{W_i\}$, $\{R_i\}$, **L**) (Fig. 1f). For practical implementation, we added an additional constraint (4.0 $V_a > V > 1.0 \, V_a$, where $V_a$ is the volume sum of compositional atoms) to avoid the generation of unreasonable structures with extremely small or large volumes.

**Optimization algorithms.** CSP is an optimization problem to identify $S$, $\{W_i\}$, $\{R_i\}$, and **L** at a given chemical composition $\{v_i\}$ to minimize $\Delta H$. After constructing crystal structure Crys($\{v_i\}$, $S$, $\{W_i\}$, $\{R_i\}$, **L**) (Fig. 1f), a structural analysis is performed and convert Crys($\{v_i\}$, $S$, $\{W_i\}$, $\{R_i\}$, **L**) to crystal graph G($\{v_i\}$,$\{e_k\}$) (Fig. 1g) and its formation enthalpy is obtained by GN model $\Delta H = f [G(\{v_i\},\{e_{ij}\})]$ (Fig. 1h).

Ideally, if one could enumerate all possible crystal structure Crys($\{v_i\}$, $S$, $\{W_i\}$, $\{R_i\}$, **L**), do crystal graph conversions to G($\{v_i\}$,$\{e_k\}$), obtain their formation enthalpy by GN model $\Delta H = f [G(\{v_i\},\{e_k\})]$, the problem of CSP was simply solved by choosing the crystal structures with the lowest $\Delta H$. However, the enumeration of all possible structures is a long-standing challenge. Here, we adopted three OAs (Fig. 1i), RAS, PSO, and BO, since

RAS and PSO are successful algorithms applied in DFT-based CSP[5,12] and BO has been shown to be compatible with the *black-box* ML model, demonstrating a great capability to identify the global minimum[34,35], and has been recently combined with DFT calculations for CSP in fixed crystal systems[36]. Here, we applied BO via a Gaussian mixture model based on the tree of Parzen estimators (TPE)[37] to explore the structural space. Compared to the normal BO algorithm based on the Gaussian process, which performs better in low-dimensional space (number of features <20), the TPE-based Gaussian mixture model demonstrated higher efficiency in high-dimensional space[37].

As shown in the right panel of Fig. 1, for a given number of atoms in a cell, $n$ initial structures Crys($\{v_i\}$, $S$, $\{W_i\}$, $\{R_i\}$, $L$) were randomly generated, and their corresponding elemental and pair attributes were obtained by structural analysis to convert the crystal structure to crystal graph G($\{v_i\}$,$\{e_j\}$). Accordingly, $\Delta H$'s were predicted using the GN model to obtain $n$ pairs of (Crys, $\Delta H_{\text{Crys}}$). After that, the approach will iteratively go the loop of structural searching from Fig. 1f–i and then back to 1f by OA. Different OAs will generate new structures in Fig. 1i, f in different ways. For RAS, the new structure was generated in a stochastic way and did not depend on the searching history. For PSO, in each iteration, a set of $n$ structures were generated as a new generation by tracking two extremes (Crys, $\Delta H_{\text{Crys}}$) values (*p*best and *g*best)[38]. We used scikit-opt (https://github.com/guofei9987/scikit-opt) and choose the momentum parameter ω as 0.8, the cognitive and social parameters are 0.5 and 0.5, respectively. For BO, a new structure with potentially low $\Delta H$ was recommended and the recommendation model was re-trained based on all previous pairs of (Crys, $\Delta H_{\text{Crys}}$) in a manner of active learning. We employ TPE-based BO as implemented in Hyperopt[37], (https://github.com/hyperopt/hyperopt) and choose observation quantile γ as 0.25[34] and a maximum number of trails to 200.

**Applications**. The GN-OA approach was then applied to identifying the crystal structures of 29 compounds listed in Table 1. There are more than 300 types of prototype structures for AB compounds[28]; two representatives of these are tetrahedral-coordination ZB/WZ and octahedral-coordination RS structures. Predicting ZB/WZ and RS structures proves the ability of CSP from ionic to covalent systems[39].

As aforementioned, the framework of approach is flexible that we adopted OQMD and MatB respectively to train GN model and RAS, PSO, and BO for the optimization algorithm. Here, we take CaS for example, to compare the performance of RAS, PSO and BO on CSP with GN model trained on MatB. The characteristics of three OAs can be clearly seen in the evolution of $\Delta H$ on the iteration steps in Fig. 3a. The $\Delta H$ distributes randomly in energy scale (Y-axis in Fig. 3a) for RAS. While, PSO can quickly find the low-$\Delta H$ configurations (exploitation). But its problem is that it may be stuck in the local minimum as shown that most of the PSO-selected structures after 1500 steps are close to each other and located around the energy of local minimum, as shown by a sharp DOS (density of structures) at a local minimum. In contrast, BO is an algorithm that has a balance between exploitation and exploration, as shown by double peaks in DOS, indicating that it has a higher ability to jump out of one particular local minimum (exploration). In this case, GN(MatB)-RAS and GN(MatB)-BO find the correct GSS at the 2503th and 372th iteration step, respectively, while GN(MatB)-PSO cannot find correct GSS within 5000 steps. For GN(MatB)-BO, the GSS was found at 207th step (Fig. 3f) with a lattice constant of 6.50 Å and then the GN(MatB)-BO show ability to optimize the lattice constant to 5.77 Å as shown in Fig. 1g, close to 5.72 Å of DFT-calculated value.

The approaches of GN-RAS, GN-PSO, and GN-BO were then applied to CSP for 28 other compounds. The results are summarized in Table 1. It was observed that: (i) like the case shown for CaS, the accuracy of OA for CSP follow the sequence that BO > RAS > PSO, whether the GN is trained on OQMD or MatB; (ii) GN model trained on MatB generally show better accuracy for CSP than that trained on OQMD, whether RAS, PSO or BO was adopted. As a result, GN(MatB)-BO shows the best performance. The corresponding $\Delta H$ evolution of GN(MatB)-RAS, GN(MatB)-PSO, and GN(MatB)-BO for all 29 compounds are shown in Supplementary Figs. 3–10. For 25 compounds that GN(MatB)-BO can correctly predict, GN(MatB)-BO can predict their lattice constants and absolute energy differences ($|\Delta H_{\text{DFT}} - \Delta H_{\text{GN}}|$) with averaged 2.24% error and 20.8 meV/atom, respectively, to DFT-calculated values, as shown in Fig. 4.

**In comparison to DFT-based approach**. Accuracy and efficiency are two criteria for a CSP approach. It should be noted that the accuracy of the current GN-OA approach is inferior to that of the DFT-based approach in terms of non-100% prediction accuracy and the variation of lattice parameters. In fact, the GN model is trained based on the DFT-calculated data; thus, it cannot surpass the accuracy of DFT results. In compromise with the accuracy, GN(MatB)-BO finished those tasks with much higher efficiency than DFT-based CSP, as shown in Fig. 5. Here, we compare the computational cost of DFT-PSO and GN(MatB)-BO to predict 25 compounds and found that GN(MatB)-BO has a computational cost three orders of magnitude lower than DFT-based approach. DFT-PSO typically requires 60–80 DFT calculations (Si and CsPbI₃ as the example shown in Supplementary Fig. 11) to find the GSS, which is consistent with previous reports of 70 and 120 DFT structural optimizations to find the GSS of GaAs (eight atoms in the cell)[10] and SiO₂ (six atoms in the cell)[12], respectively.

### Discussion

To the best of our knowledge, this is the first study to establish a GN-OA framework for CSP, which contains three essential parts: (1) a database consisting of crystal structures and the formation energies; (2) a GN constructing the correlation model between crystal structure and formation energies; (3) an OA to search the crystal structures with minimum formation energy. These three parts are all fast-developing research frontiers and certainly not perfect at present; therefore, the limitations of the current GN-OA approach are also apparent, such as the failure to predict the GSS of some crystals and the deviation of predicted lattice parameters. There are two failure modes. One is the failure of GN model to put the GSS as the lowest $\Delta H$, such as CdS (Supplementary Fig. 12), and the other is the failure of OA not visit the GSS with the lowest $\Delta H$, such as GN-PSO for CaS (Supplementary Fig. 13). Meanwhile, their advantage is that any progress of these three aspects may help in improving the efficiency and accuracy of GN-OA approach.

In this study, we adopted and compared OQMD and MatB databases, mainly containing stable or metastable structures (global or local minimums in PES). However, during the structural searching process, most structures are unstable (away from the minimums). The addition of the energetic data of these unstable structures should help the model to capture the entire PES landscape, thereby improving the efficiency and accuracy of GN-OA approach. Notably, generating energy landscape on numerous unstable structures is a necessary step for generating ML potential[40]. In principle, CSP based on ML potential should be more accurate; however, ML potential is generated on fixed

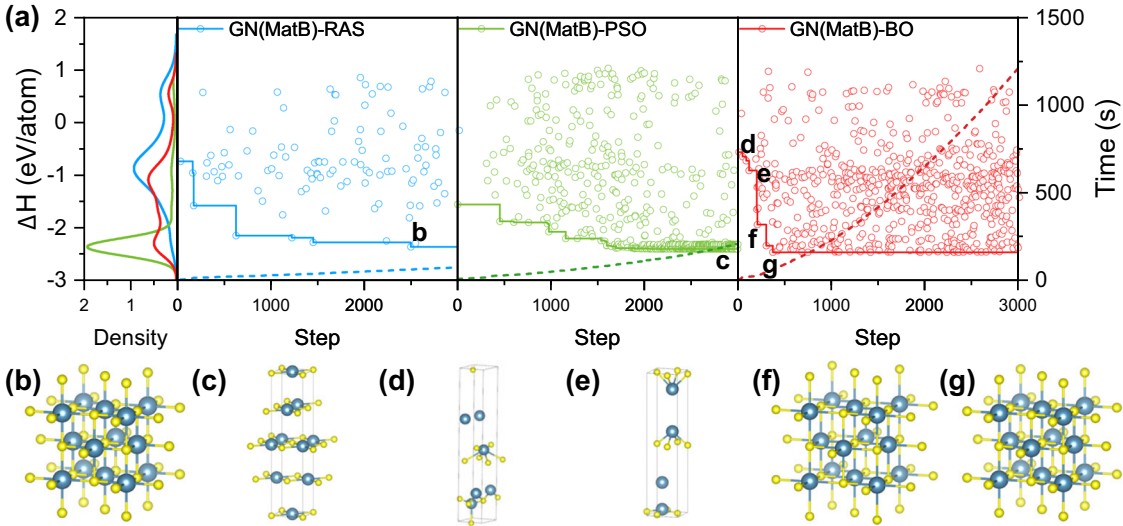

**Fig. 3 The process and performance of GN-OA. a** Process of GN(MatB)-RAS, GN(MatB)-PSO, and GN(MatB)-BO approaches, to search the crystal structure of CaS. The density of structures (DOS) at the energy level has been shown in the left panel. The computational cost has been added as the dashed lines. **b–g** Typical structures in the above process as shown in (**a**).

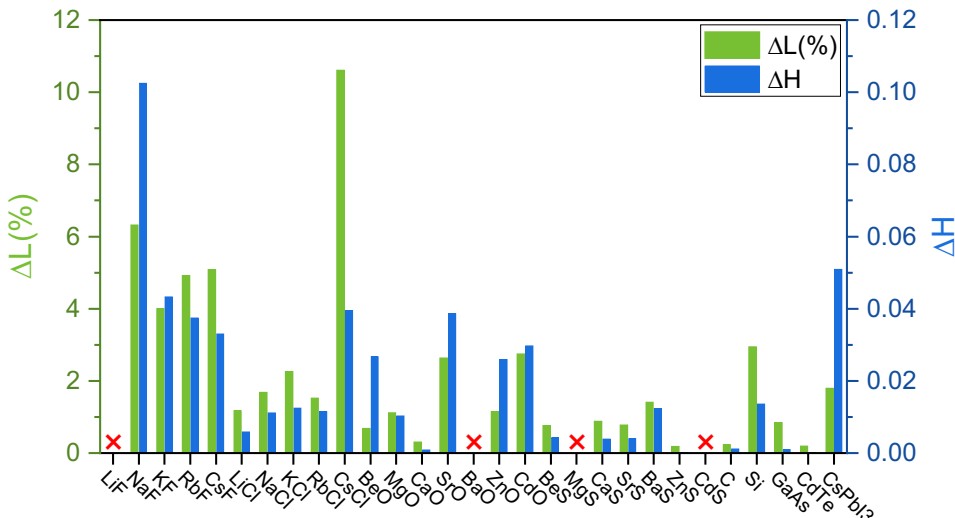

**Fig. 4 The comparisons of GSS derived from GN-OA and DFT.** The discrepancies of lattice constants and formation energies for 29 compounds predicted by GN(MatB)-BO approach in comparison to DFT-calculated results. The red cross mark means that GN(MatB)-BO predicted wrong GSS in comparison to DFT results.

types of elemental combinations [{$v_i$} is constant], for example of aluminum[40], while the GN model for CSP should be universal for all elements [{$v_i$} is a variable]. For constant {$v_i$}, 6000–10,000 DFT calculations are required to generate an applicable ML potential. It is an open question that how many DFT data are required to generate a reasonable GN model and how to combine existing DFT data trained for ML potential generation into GN model. This requires further investigation.

n this study, we adopted the framework of MEGNet[19] as a crystal graph. Since the first development of crystal graph (CGCNN)[18], many studies are being conducted to further improve the crystal graph, such as improved crystal graph convolutional neural network (iCGCNN)[41], directional message passing neural network (DimeNet)[42], atomistic line graph neural network (ALIGNN)[19,43–47], which was reviewed in a recent paper[48]. The implementation of those crystal graphs in GN-OA framework or further development of crystal graphs may further improve the accuracy.

We show that BO algorithm when combined with GN model is superior to PSO and RAS, which are often combined with DFT for CSP. Notably, BO is also replaceable. An optimization algorithm that is compatible with black-box GN model needs further exploration.

A platform will be established to allow the users to combine their crystal representation, database, and structural searching approach to optimize GN-OA approach for CSP. In addition to the database, crystal graph, and optimization algorithm, opportunities are given to technical improvements, such as algorithm parallelization and optimization, which may also improve the accuracy and efficiency.

In summary, we constructed a flexible framework that used a graph network to establish the ML model between crystal structures and their formation enthalpies at the given database, and this model was then combined with an optimization algorithm for CSP. The framework was then applied to predict the crystal structures of 29 typical compounds. The comparative studies of

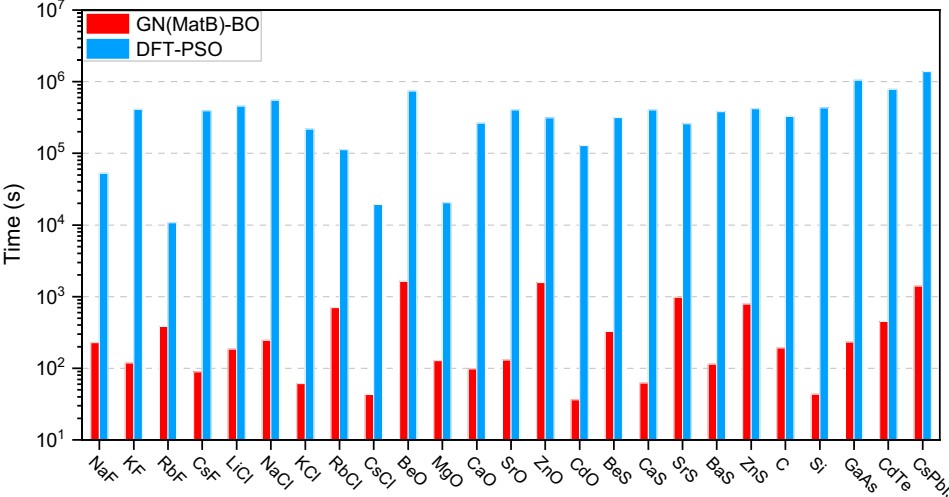

**Fig. 5 The comparison of computational cost.** The computational cost of GN(MatB)-BO and DFT-PSO to find the right crystal structures. The computational time is accounted for running on CPU core [Intel(R) Xeon(R) Silver 4210 CPU @ 2.20 GHz].

multiple combinations of database, GN model, and optimization algorithm showed that GN model trained on MatB combined with Bayesian optimization structural searching [GN(MatB)-BO], although with less accuracy than DFT results, can predict crystal structures with computational cost three orders of magnitude less than DFT-based approaches. Meanwhile, the limitations of the current GN-OA approach are also apparent. In terms of methodology, several directions need further development, including crystal structure characterization, structural searching, and algorithm parallelization, to predict more complicated and unknown structures more efficiently. The current study may open a new avenue for data-driven crystal structural prediction without using the expensive DFT calculations during structural searching.

## Data availability
All relevant data are included in this article and its Supplementary Information files.

## Code availability
The code for GN-OA approach is available on http://www.comates.group/links?software=gn_oa. All GN-based results reported in this work can be reproduced by this code. The DFT-based results are produced by CALYPSO code.

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

## Acknowledgements

W.Y. acknowledged funding support by National Key Research and Development Program of China (Grant No. 2020YFB1506400), National Natural Science Foundation of China (Grant No. 11974257), Jiangsu Distinguished Young Talent Funding (Grant No. BK20200003), Yunnan Provincial Key S&T Program (Grant No. 202002AB080001-1), the Priority Academic Program Development of Jiangsu Higher Education Institutions (PAPD). DFT calculations were carried out at the National Supercomputer Center in Tianjin [TianHe-1(A)].

## Author contributions

W.Y. conceived the idea. G.C. wrote the code and conducted the calculations. X.G., G.C., and W.Y. discussed the results and wrote the manuscript.

## Competing interests

The authors declare no competing interests.
