## [Peer Review File · Nature Communications]

REVIEWER COMMENTS

Reviewer #1 (Remarks to the Author):

The authors propose a new algorithm and approach for predicting a material property (formation enthalpy) of materials based on data science techniques. The approach leverages both graph network representations and bayesian optimization. I found the structure and flow of the paper to be acceptable but cannot recommend this paper for publication for the following reasons:

1. While I was able to download and run an example from the link provided by the authors, I noticed that there was no code. There were only .pyc files. This is not acceptable. Best practices in materials informatics require full data and code in order to reproduce work.

2. I found the methods section insufficient. Greater detail is necessary in order for future authors to follow and reproduce the work. In particular, the feature subsection was inadequate. The caption for figure 5 could be more detailed.

3. I was also not impressed with the scope and benchmarks provided. Formation enthalpy is a very widely studied property for materials informatics papers. The authors make comparisons with MEGNet, iCGCNN etc, but are not using benchmark datasets. This is not acceptable. Small changes in the dataset, splitting used, cleaning method used etc can lead to differences that make error comparisons meaningless. The authors are encouraged to use benchmark datasets. For example <https://doi.org/10.1038/s41524-020-00406-3>

Reviewer #2 (Remarks to the Author):

The problem addressed by this work is a fundamentally important one. Conceptually, the approach makes sense and seems to combine several good technical ingredients.

However, the exposition has some major issues such that I can not yet adequately evaluate the quality of the experimental design and results. It seems likely to me that there are gaps in the study design. These issues are

#1 Number of atoms in unit cell

A fundamental challenge in CSP is the growth in potential atomic arrangements as you add more atoms to the unit cell and it is not known a priori how many atoms are in the ground state unit cell. All of the exposition of the approach suggests a fixed number of atoms are used. However, the various figures seem to show unit cells with different numbers of atoms. This must be explained.

#2 Training data

First, the manuscript in various places claims the method is “without using expensive DFT calculations”. This is simply false. The GN is trained (in part) on DFT calculations. I believe the authors meant to make a more precise statement about *when* DFT calculations are used that would be true.

Second, it is not clear what training data is used for the GN. Lines 82-91 suggest that there was no training data that included any of the atomic species that are used in GN-BOSS. However, I believe Fig 2 includes data with those atomic species. This needs to be clarified and performance data about the actual GN model used during the BO (not a related model trained on different data (which is what I am guessing is being shown)). If you are getting generalization to atomic species never observed in training such that the model is precise enough for CSP, that would be a surprising and interesting result.

#3 Clarity in the BO procedure

It is not clear what the statement “GN-BO can find the ground state structures” actually means. I think this should mean that the structure with the lowest energy predicted by the GN among everything considered in the entire BO trajectory is close to the ground state. However, it could mean that the authors chose structures explored by the procedure that were close to the ground state regardless of the predicted energy. These are obviously vastly different claims. The actual procedure needs to be accurately described.

#4 Lack of quantitative measurements

The progression / closeness of a structure to the true ground state is purely qualitative (e.g. “Fig 3(e) still has some distortions”) with a few pictures. A quantitative way to establish the closeness of a structure to the ground state must be established.

Notably, the claim on line 163 I can not observe given the figure.

#5 Lack of model clarity

The graph model is not adequately described. The addition of voronoi polyhedra features is a sound approach, but the reader is not given details of how that is done. Fig 5 suggests there are two different Voronoi tessellations being done (for atoms and edges), but this needs to be described precisely, including how unit cell edges are handled.

Overall, the architecture of the GN model as described in lines 242-244 and Fig 6 is inadequate to allow the reader to reproduce the work. Further, the training procedure is not described. Note that line 101 claims the details are in the Supplementary Materials when they are not.

Similarly, the TPE approach has a number of hyperparameters that are never described.

#6 -SYM comparisons

From looking at Figured S5-S9, I believe the GN-BO largely doesn't find ground state structure. Only GN-BO-SYM finds them (though comment #4 may mean I misunderstand the pictures). This leaves open the possibility that the only important part of the algorithm is implementing the -SYM constraints. These experiments do not establish that the GN plus the TPE based acquisition function are important for achieving the given results.

Also, the entire description of these constraints is in line 185, which essentially says nothing about what they are. This seems to be an important technical point and must be described.

Fundamentally, this work introduces 3 separate things:

1. A graph network with some new features to be the proxy model for DFT in CSP.
2. An acquisition function based on TPE in a BO for CSP.
3. An acquisition function set of symmetry constraints.

The experiments should establish which of these are important for the functioning of the system. Notably, (3) may (I can't tell because these constraints are not explained in the text) severely limit the applicability of the method because it may require a great deal of information about your target structure.

I have a significant number of smaller comments on the manuscript, but given the large issues above, I will wait until a potential follow on review before going into those.

Lastly, I have two comments for the authors that I would not necessarily consider part of refining this work, but I think may be useful in the future.

1. There is a body of work in using ML proxies for DFL calculations in the force field / molecular dynamics literature. There are useful ideas, especially around when to run the DFT calculations if the model is uncertain. This paper is a good example of this:

Smith, Justin S., Benjamin Nebgen, Nithin Mathew, Jie Chen, Nicholas Lubbers, Leonid Burakovsky, Sergei Tretiak, et al. 2021. "Automated Discovery of a Robust Interatomic Potential for Aluminum." *Nature Communications* 12 (1): 1257.

2. I note that outside of the -SYM cases, symmetries and invariances may not be given to the model in any way. For example, rotations of the unit cell will look like different inputs to the model. Often, making these symmetries/ invariances structurally available to the model (meaning that it doesn't have to learn them from data) or using data augmentation would be worth considering in the future.

Reviewer #3 (Remarks to the Author):

In this paper, the authors address the problem of crystal structure prediction. Rather than using DFT, they train a graph neural network surrogate energy function on data from the Open Quantum Materials Database. Being able to solve crystal structures without DFT would be a major accomplishment, but I am not convinced that the paper accomplishes this in a convincing way, on a sufficiently hard problem.

As such I think the paper is fit for publication in its current form. In particular I have concerns about the preparation of the training set, the analysis provided in the paper and the fact that statements about computational costs seem exaggerated.

My main issue with the paper is that there is only very poor comparison with existing methodology. I have little personal experience of crystal structure determination with ab initio random structure search or similar methods, but my understanding is that it is often easy to find structures with small number of optimizations, for example consider fig. 7 from the AIRSS paper (<https://arxiv.org/pdf/1101.3987.pdf>). This figure is showing that for unit cells with less than 20 atoms, it is often possible to find a structure with a handful of DFT optimizations. I would like the authors to compare their method with existing methods from the literature, clearly stating what the mean number of DFT evaluation required to find the minimum are for each method.

I also have some specific questions that need addressing:

1. Did the 24 binary complexes were not included in the training set? The text is ambiguous, could the authors make it crystal clear that models were neither trained nor selected (validated) based on their performance on the 24 binary complexes or on the carbon, silicon or CsPbI₃ discussed in the supplemental information. Using such data in the training set would pollute the conclusions.

2. Fig S2. vs Fig 2. The errors on the RAS are still much larger than on the training set. This seems to suggest that the GN are very overfit on non-equilibrium structures. Does this not detract from our trust in the models? Would it be possible to show the results in S2 in the same way as Fig 2 to show how the model performs far from the training set.

3. Fig 3: why does GN-BO-SYM immediately find the minimum? It looks to me in figures S5-S10 that GN-BO and GN-BO-SYM often (always) produce different structures and that the structures that include symmetry information always look more sensible. Could symmetry information also be used in the random search or particle swarm approaches? In fact, how exactly is the symmetry method implemented. I am guessing the authors are using information about the point group of the ground truth structure, but this should be stated explicitly.

4. I find it difficult, from the materials included in the paper, to determine how close the structures predicted by the different methods differ from the DFT ones for the different models. From Fig 3, it looks like only GN-BO-SYM is able to find the same minimum as a DFT based method. From Figs S5-S10 I am not sure what the DFT structures would be. Also the shifting of the energy scales makes it difficult to assess how different those optimized structures really are in terms of the energy evaluation from the GN. Ideally it would be good to know what the ΔH values that each optimization method reaches according to the graph net model and what differences with respect to the 'true' DFT structure this translates into when evaluated with DFT itself.

5. Does Figure 4 contain the cost of a single DFT optimization to be performed after each GN-BO run? The main text states that 'We further used DFT calculations to relax the GN-BO 176 predicted structures and obtain the precise crystal structures.' . In fact the statement that the computational cost is three orders of magnitude lower than for a DFT based method seems an exaggeration. For example, in the case of CaS the authors show simply running DFT on 18 canonical starting structures, leads to the correct structure being determined three times out of 18. This means that one would expect for the cost to be approximately 3 DFT optimizations to find this crystal structure. On the other hand the GN-BOSS method performs at least a single DFT optimization. Could the authors clarify this point?

There are smaller issues to do with clarity and the possibility to replicate the authors work, but these are not important unless the authors can show that the points above can be resolved.

REVIEWER COMMENTS

Reviewer #1 (Remarks to the Author):

The authors propose a new algorithm and approach for predicting a material property (formation enthalpy) of materials based on data science techniques. The approach leverages both graph network representations and bayesian optimization. I found the structure and flow of the paper to be acceptable but cannot recommend this paper for publication for the following reasons:

Response: We appreciate the reviewer's assessment of our work.

1. While I was able to download and run an example from the link provided by the authors, I noticed that there was no code. There were only .pyc files. This is not acceptable. Best practices in materials informatics require full data and code in order to reproduce work.

Response: We updated the source code in the 'Code Availability' section. An example for predicting the crystal structure of CaS has been provided as the directory of './example/CaS'.

The code will be open sourced on Github after the publication of the paper.

2. I found the methods section insufficient. Greater detail is necessary in order for future authors to follow and reproduce the work. In particular, the feature subsection was inadequate. The caption for figure 5 could be more detailed.

Response: We have expanded the description of the approach and merged the original Methods section into the main text for clarity. Currently, the Results section is reorganized as follows:

- Crystal Graph
- Database and data split
- GN model
- Symmetry constraint
- Optimization algorithms
- Applications
- In comparison to DFT-based approach
- Limitations and future work.

Regarding the reviewer's comment "*In particular, the feature subsection was inadequate*". The atomic and bond features were depicted by N_v - and N_e - dimensional vectors (N_v and N_e are hyperparameters) learned from model training. N_v -dimensional vector is the embedded atomic number of element, and N_e -dimensional vector is the Gaussian-expanded distance of the bond. The details of crystal graph have been included under the section 'Results/Crystal graph'. For better flow, Figure 1 was revised to incorporate Figures 5 and 6 in original version.

3. I was also not impressed with the scope and benchmarks provided. Formation enthalpy is a very widely studied property for materials informatics papers. The authors make comparisons with MEGNet, iCGCNN etc, but are not using benchmark datasets. This is not acceptable. Small changes in the dataset, splitting used, cleaning method used etc can lead to differences that make error comparisons meaningless. The authors are encouraged to use benchmark datasets. For example <https://doi.org/10.1038/s41524-020-00406-3>

Response: We agree with the reviewer that the direct comparison of mean absolute error (MAE) among different models is meaningless, owing to the variation of the dataset and data split in literature. In the revised manuscript, we have omitted the comparison between our MAE with those in previous studies that adopted different datasets or data splits. We have expanded our study to two benchmark datasets, *i.e.*, OQMD (offline version 1.3) [JOM 65, 1501 (2013)] and Matbench dataset of formation energy (MatB) [npj Computational Materials 6, 138 (2020)]. The results of the GN model applied on two types of databases (OQMD and MatB) with the same data split (training/validation/test ratio: 4/1/3) are shown in the revised Figure 2.

We have revised section "Results/Database and data split" as follows for clarity:

“Database and data split. Two benchmark datasets, OQMD of version 1.3²⁸, and MatB²⁹ have been used for GN model training and evaluation. For OQMD, data cleaning was performed to exclude data with incomplete information and restrictions: i) the number of atoms in the unit cell (less than 50), ii) PBE as exchange-correlation functional, and iii) kinetic energy cutoff (520 eV), making data as reliable and comparable as possible. Accordingly, more than 320,000 data points have been obtained, including ~40,000 experimentally known ones and ~280,000 hypothetical ones, covering 85 elements, 7 lattice systems, and 167 space groups. For MatB, we used the Matbench v0.1 dataset²⁹ that is derived from data cleaning in Materials Project. For properties of formation energy, it included ~132,000 data points, covering 84 elements, 7 lattice systems, and 227 space groups. The distributions of the number of elements and atoms in each database are shown in Figures 1(a) and (b). For both OQMD and MatB, the same ratio of data split has been adopted, i.e., training set (50%), validation set (12.5%), and test set (37.5%), to construct GN models for CSP. In all the training, validation and test process, the data of 29 binary compounds studied in this work, have been excluded.”

The GN models trained on OQMD and MatB have been compared under “Results/GN model”, as follows:

“The results show that GN(OQMD) has less MAE (16.07 meV/atom) than GN(MatB) (31.66 meV/atom). MAE of GN(MatB) is close to the previous report of 32.7 meV/atom²⁹. Such a tiny difference of 1 meV on the same MatB dataset may originate from different data split. The insets in Figure 2 show a systematic decrease of the MAE as the number of training data. Better performance of OQMD can be ascribed to its larger database (~ 320000 DFT-calculated data for inorganic compounds), which is more than twice than MatB. Despite less MAE of GN(OQMD), as shown later, its performance on CSP is inferior to GN(MatB), indicating possible overfitting of GN(OQMD).”

We hope that we have satisfactorily addressed all of the reviewer’s comments. We will be happy to provide more clarification, if required.

Reviewer #2 (Remarks to the Author):

The problem addressed by this work is a fundamentally important one. Conceptually, the approach makes sense and seems to combine several good technical ingredients.

However, the exposition has some major issues such that I can not yet adequately evaluate the quality of the experimental design and results. It seems likely to me that there are gaps in the study design. These issues are

Response: We thank reviewer for assessing our work and for acknowledging the importance of the topic. We apologize for the confusion and ambiguousness in the previous version. In the revised manuscript, we have accordingly improved the algorithm, reorganized the content,

and clarified details about the algorithm. The substantial changes include:

- **Improvement of algorithm**

1. Flexible framework of the algorithm

Inspired by the comments from this Reviewer, the proposed algorithm has been summarized under three essential sections: a database, a graph network (GN), and an optimization algorithm (OA). The contents are accordingly reorganized, and an additional database, MatB, has been implemented for GN model training.

2. Enforced symmetry constraint

Crystal structural prediction without symmetry constraint (CSP-noSYM) leads to many issues, such as the unwanted distortions and last-step DFT relaxation to close a structure, which have been raised by Reviewer #2 and #3. Considering the fact that most of crystal structures has symmetry operations, CSP-noSYM is unnecessary. In the revised manuscript, a symmetry constraint is enforced for CSP and the corresponding implementation details have been explained in the section ‘Results/Symmetry constraint’.

3. Crystal graph

In original version, the crystal graph was depicted by 23 fixed atomic features and 4 structural features based on Voronoi polyhedron. In the revised manuscript, following the MEGNet [Nature Computational Science 2021, 1, 46; Chem. Mater. 2019, 31, 3564], the crystal graph was depicted by N_v - and N_e - dimensional vectors (N_v and N_e are hyperparameters) learned from model training. N_v -dimensional vector is the embedded atomic number of element, and N_e -dimensional vector is the Gaussian-expanded distance of the bond. The details of crystal graph have been included under the section ‘Results/Crystal graph’.

- **Reorganization of content**

1. We have expanded the description of the approach and merged the original Methods section into the main text for clarity. Currently, the Results section is reorganized as follows:

- Crystal Graph
- Database and data split
- GN model
- Symmetry constraint
- Optimization algorithms
- Applications
- In comparison to DFT-based approach
- Limitations and future work

2. Figure 1 was elaborated on to convey a clear and complete picture on how the approach works.

3. We take 8-atom cell for the CSP of all compounds except CsPbI₃ (10-atom cell) to avoid confusing the readers with a variable number of atoms in different figures, as commented by this Reviewer.

We address reviewer's concerns point-by-point below.

#1 Number of atoms in unit cell

A fundamental challenge in CSP is the growth in potential atomic arrangements as you add more atoms to the unit cell and it is not known a priori how many atoms are in the ground state unit cell. All of the exposition of the approach suggests a fixed number of atoms are used. However, the various figures seem to show unit cells with different numbers of atoms. This must be explained.

Response: The reviewer is correct that in CSP, one should NOT know a priori how many atoms are in the ground state unit cell. What the computation does is searching for the stable structure when the number of atoms in the cell is fixed, *i.e.*, minimizing $\Delta H = f(\text{Crys}(\{\mathbf{v}_i\}_{i=1,N}, \{\mathbf{R}_i\}_{i=1,N}, \mathbf{L}))$ on $(\{\mathbf{R}_i\}_{i=1,N}, \mathbf{L})$ at given $\{\mathbf{v}_i\}_{i=1,N}$.

Regarding the reviewer's comment "*the various figures seem to show unit cells with different numbers of atoms*", in the revised manuscript, we are considering 8-atom cell for CSP for all compounds, except for CsPbI₃ (10-atom cell) to avoid confusion.

#2 Training data

First, the manuscript in various places claims the method is "without using expensive DFT calculations". This is simply false. The GN is trained (in part) on DFT calculations. I believe the authors meant to make a more precise statement about *when* DFT calculations are used that would be true.

Response: We fully understand the reviewer's concerns and have improved our tone to remove the words 'without DFT' and 'DFT-free' in the revised manuscript.

Second, it is not clear what training data is used for the GN. Lines 82-91 suggest that there was no training data that included any of the atomic species that are used in GN-BOSS. However, I believe Fig 2 includes data with those atomic species. This needs to be clarified and performance data about the actual GN model used during the BO (not a related model trained on different data (which is what I am guessing is being shown)).

Response: The GN model did include information on atomic species. To clarify the algorithm, we have expanded Figure 1 to include the necessary information into one figure. We have used atomic number embedding, as proposed in MEGNet [Nature Computational Science 2021, 1, 46; Chem. Mater. 2019, 31, 3564]. In the revised manuscript, the following statement has been added in the section "Results/Crystal graph":

*"Crystal graph. In the original GN³⁰, a graph is defined by three ingredients, *i.e.*, nodes (\mathbf{v}_i),*

edges connecting nodes (e_k), and the global attributes (\mathbf{u}), which are naturally borrowed to crystal graph as atoms, bonds, and macroscopic attributes (e.g., pressure, temperature)^{19,31}. Considering that multiple atoms and bonds exist in a crystal, crystal graph is numerically represented by $G(\{\mathbf{v}_i\}_{i=1:nv}, \{\mathbf{e}_k\}_{k=1:ne}, \mathbf{u})$, where \mathbf{v}_i and \mathbf{e}_k are the elemental and bond attributes of i th atom and k th bond, and n_v and n_k are the number of atoms and bonds, respectively, in the cell. In MEGNet¹⁹, v and e are the atomic numbers and spatial distance, represented by N_v - and N_e - dimensional vectors (N_v and N_e are hyperparameters) learned from model training, respectively. Accordingly, an embedding layer with a $N_v \times n_v$ matrix [Figure 1(c)] was added after atomic attribute $\{\mathbf{v}_i\}$ as input for GN [Figure 1(j)]. A $n_v \times n_v \times N_e$ matrix [Figure 1(d)] was added after $\{\mathbf{e}_k\}$ [Figure 1(k)], where $n_v \times n_v$ represents the bond connectivity between two atoms and each bond is represented by an expanded distance with Gaussian basis numerically represented by N_e points. In comparison to the fixed features, N_v - and N_e - dimensional vectors can be considered as elemental and bond features that were learned during the model training process. The learned elemental embeddings have been shown to encode the elemental periodicity and can be transferred to predict different properties¹⁹.”

If you are getting generalization to atomic species never observed in training such that the model is precise enough for CSP, that would be a surprising and interesting result.

Response: We appreciate these insightful and interesting comments. We performed two tests. One involved deleting all crystal data that include element C in MatB, training the GN model, and using GN(MatB)-BO to predict a stable structure of 8-atom C. It is surprising that GN(MatB)-BO determines the diamond structure as the most stable one, as shown in Figure R1(a-c).

The other test involved deleting all crystal data that include element Si in MatB, training GN model, and using GN(MatB)-BO to predict a stable structure of 8-atom Si. Unfortunately, GN(MatB)-BO predicted a wrong ground state structure, as shown in Figure R1(d-f). Meanwhile, the stable diamond structures appear frequently at an energy of approximately 0.50 eV/atom above, as shown in Figure R1(e) and R1(g), indicating that a stable diamond structure could be a local minimum in the GN model.

Figure R1: Elemental appearance in MatB excluding all crystals involving element of (a) C

and (d) Si. The results of crystal structure prediction by GN(MatB)-BO for (b-c) C crystals via GN model trained without C species and for (e-g) Si crystals via GN model trained without Si species.

Based on the above tests, we could see that the current approach shows potential for CSP atomic species never observed in training, but is still not perfect for all cases. This could be improved from three aspects, adding more representative data in the database, and improving the GN model. We have discussed these issues in the section ‘Limitations and future work’ in the revised manuscript.

#3 Clarity in the BO procedure

It is not clear what the statement “GN-BO can find the ground state structures” actually means. I think this should mean that the structure with the lowest energy predicted by the GN among everything considered in the entire BO trajectory is close to the ground state. However, it could mean that the authors chose structures explored by the procedure that were close to the ground state regardless of the predicted energy. These are obviously vastly different claims. The actual procedure needs to be accurately described.

Response: It means “the structure with the lowest energy predicted by the GN among everything considered in the entire BO trajectory is close to the ground state”.

#4 Lack of quantitative measurements

The progression / closeness of a structure to the true ground state is purely qualitative (e.g. “Fig 3(e) still has some distortions”) with a few pictures. A quantitative way to establish the closeness of a structure to the ground state must be established. Notably, the claim on line 163 I can not observe given the figure.

Response: Thank you very much for these insightful comments. It is true that without the symmetry constraint, there are challenges to accurately predict a perfect structure without distortion in the previous GN model. In the original version, we used DFT relaxation in the final step to close a structure.

Considering the fact that most crystal structures have symmetry, the application of CSP without the symmetry constraint results in unnecessary challenges such as unwanted distortions, last-step DFT relaxation, and difficulty to claim a closeness to find a structure. Therefore, in the revised manuscript, the symmetry constraint is enforced. The problem of unwanted distortions has been solved, and the last-step DFT relaxation is not yet necessary. In symmetry constraint, we claim a closeness of GSS that the decrease of ΔH is less than 10 meV/atom within 500 iteration steps.

Regarding the reviewer’s comment, “Notably, the claim on line 163 I can not observe given the figure.”, the claim in line 163 that “GN-BO shows superior optimization performance than that of GN-PSO or GN-RAS” is based on the BO step required for each method to find the correct ground state structure (GSS). GN-RAS and GN-PSO do not find the correct GSS

within 6000 steps, as shown in the original Figure 3f and 3g, respectively. GN-BO and GN-BO-SYM required 4,047 and 254 steps to find the correct GSS, as shown in the original Figure 3e and 3h. Such comparisons among approaches without the symmetry constraint have been deleted, since the symmetry constraint is now enforced in revision.

#5 Lack of model clarity

The graph model is not adequately described. The addition of voronoi polyhedra features is a sound approach, but the reader is not given details of how that is done. Fig 5 suggests there are two different Voronoi tessellations being done (for atoms and edges), but this needs to be described precisely, including how unit cell edges are handled.

Overall, the architecture of the GN model as described in lines 242-244 and Fig 6 is inadequate to allow the reader to reproduce the work. Further, the training procedure is not described. Note that line 101 claims the details are in the Supplementary Materials when they are not. Similarly, the TPE approach has a number of hyperparameters that are never described.

Response: In the original version, the graph model of the crystal consists of two parts: (1) atoms and (2) edges (atomic bonding or connection). Atoms are represented by 23 atomic features $\{\mathbf{v}_i\}$, and edges are represented by four features $\{\mathbf{e}_k\}$ of Voronoi tessellations for edges (not atoms). In revision with symmetry constraint, we have updated the crystal graph by replacing Voronoi tessellations with Gaussian expansion of spatial distance, which shows better accuracy while predicting lattice constants.

In the revised manuscript, Figure 1 is revised to clearly show the crystal graph [Figure 1(c)(d)] and the flowchart of GN-OA approach. The graph models have been explained in section “Results/crystal graph” as follows:

“Crystal graph. In the original GN³⁰, a graph is defined by three ingredients, i.e., nodes (\mathbf{v}_i), edges connecting nodes (\mathbf{e}_k), and the global attributes (\mathbf{u}), which are naturally borrowed to crystal graph as atoms, bonds, and macroscopic attributes (e.g., pressure, temperature)^{19,31}. Considering that multiple atoms and bonds exist in a crystal, crystal graph is numerically represented by $G(\{\mathbf{v}_i\}_{i=1:nv}, \{\mathbf{e}_k\}_{k=1:ne}, \mathbf{u})$, where \mathbf{v}_i and \mathbf{e}_k are the elemental and bond attributes of i th atom and k th bond, and n_v and n_k are the number of atoms and bonds, respectively, in the cell. In MEGNet¹⁹, \mathbf{v} and \mathbf{e} are the atomic numbers and spatial distance, represented by N_v - and N_e - dimensional vectors (N_v and N_e are hyperparameters) learned from model training, respectively. Accordingly, an embedding layer with a $N_v \times n_v$ matrix [Figure 1(c)] was added after atomic attribute $\{\mathbf{v}_i\}$ as input for GN [Figure 1(j)]. A $n_v \times n_v \times N_e$ matrix [Figure 1(d)] was added after $\{\mathbf{e}_k\}$ [Figure 1(k)], where $n_v \times n_v$ represents the bond connectivity between two atoms and each bond is represented by an expanded distance with Gaussian basis numerically represented by N_e points. In comparison to the fixed features, N_v - and N_e - dimensional vectors can be considered as elemental and bond features that were learned during the model training process. The learned elemental embeddings have been shown to encode the elemental periodicity and can be transferred to predict different

*properties*¹⁹.”

Regarding the reviewer’s comment “Further, the training procedure is not described. Note that line 101 claims the details are in the Supplementary Materials when they are not.” In the revised manuscript, the following statement has been added in the section “Results/GN model”:

“GN model. As shown in Figure 1(h), the GN model was constructed to establish the correlation between the crystals and their formation enthalpies¹⁹. Crystal graph represented by matrix $\{v_i\}$ [Figure 1(c)(j)] and $\{e_k\}$ [Figure 1(d)(k)] are the input of GN model and formation enthalpy ΔH is the output. There could be m MEGNet layers (m is hyperparameter) that make up the MEGNet blocks [Figure 1(l)] to update the matrix $\{v_i\}$ and $\{e_k\}$. The set2set layers [Figure 1(m) and (n)] are used to learn a representation vector from the matrix $\{v_i\}$ and $\{e_k\}$. Then use the concatenate layer [Figure 1(o)] to combine these vectors, go through a fully connected layer [Figure 1(p)] composed of l dense layers (l is hyperparameter), and get ΔH [Figure 1(q)]. We train GN model using the data in two respective databases, i.e., OQMD²⁸, and MatB²⁹, leading to two different GN models, GN(OQMD) and GN(MatB). By optimizing hyperparameters in Table S1, the best performing one in each model was selected to minimize the errors between the GN-predicted and DFT-calculated ΔH on the test set as the results shown in Figure 2. The results show that GN(OQMD) has less MAE (16.07 meV/atom) than GN(MatB) (31.66 meV/atom). MAE of GN(MatB) is close to the previous report of 32.7 meV/atom²⁹. Such a tiny difference of 1 meV on the same MatB dataset may originate from different data split. The insets in Figure 2 show a systematic decrease of the MAE as the number of training data. Better performance of OQMD can be ascribed to its larger database (~ 320000 DFT-calculated data for inorganic compounds), which is more than twice than MatB. Despite less MAE of GN(OQMD), as shown later, its performance on CSP is inferior to GN(MatB), indicating possible overfitting of GN(OQMD).”

Regarding the reviewer’s comment, “the TPE approach has a number of hyperparameters that are never described”, we have added the statement in the section “Results/Optimization Algorithms” that “We employ TPE-based BO as implemented in Hyperopt^{37,40} and choose observation quantile γ as 0.25³⁴ and maximum number of trails to 200.” The hyperparameters used in this work are also listed in Table S1 for clarity.

#6 -SYM comparisons

From looking at Figure S5-S9, I believe the GN-BO largely doesn’t find ground state structure. Only GN-BO-SYM finds them (though comment #4 may mean I misunderstand the pictures). This leaves open the possibility that the only important part of the algorithm is implementing the -SYM constraints. These experiments do not establish that the GN plus the TPE based acquisition function are important for achieving the given results.

Response: In the previous manuscript, GN-BO did find most of the GSSs, although the structures seem different from those derived from GN-BO-SYM. Such a difference could be caused by a visual discrepancy due to *periodic invariance*, i.e., there are infinite different

ways of choosing unit cells with different shapes and sizes, as a schematic example shown in Figure R2 below. Since GN-BO has no symmetry constraint, the cell shape can vary and seems significantly different from the conventional cell derived from GN-BO-SYM. Such a problem was solved using a conventional cell (each space group has its typical conventional cell) with symmetry constraint in the revised manuscript. Symmetry constraint is important but not “*the only important part of the algorithm*”. In symmetry constraint, we did not fix the space group as prior information but consider it as a variable (from P2 to P230) during optimization. As shown in the revised manuscript, a proper database, GN model, and optimization algorithms are all important for an efficiency and accurate CSP.

Figure R2: The same two dimensional square lattice can have different unit cell shapes as shown by $(\mathbf{a}_1, \mathbf{b}_1)$, $(\mathbf{a}_2, \mathbf{b}_2)$, $(\mathbf{a}_3, \mathbf{b}_3)$, $(\mathbf{a}_4, \mathbf{b}_4)$, and $(\mathbf{a}_5, \mathbf{b}_5)$.

Also, the entire description of these constraints is in line 185, which essentially says nothing about what they are. This seems to be an important technical point and must be described.

Response: We have added a section ‘Results/Symmetry constraint’ in the revised manuscript to explicitly state how symmetry constraint has been applied.

“Symmetry constraint. The wealth of experimental data shows that most of crystal structures at low temperature have symmetry operations³² and adding symmetry constraint would accelerate CSP. Meanwhile, most crystal structures in training data, either OQMD or MatB, are symmetrical (with space group spanning from P2 to P230). In this work, we treat CSP with symmetry constraint, by adding two additional structural features, crystal symmetry S and the occupancy of Wyckoff position W_i for the i th atom, which are chosen through 229 space groups and associated 1506 Wyckoff positions³³. As shown in Figure 1(e), the procedure firstly chose a symmetry S among P2 to P230 and then the lattice parameters \mathbf{L} are generated within the chosen symmetry. Secondly, a combination of Wyckoff positions $\{W_i\}$ at given symmetry is selected to meet the number of atoms given in a cell. The atomic coordinates $\{\mathbf{R}_i\}$ are then given by the selected Wyckoff position $\{W_i\}$ and lattice parameters \mathbf{L} . Space group S and corresponding $\{W_i\}$ are variables upon optimizations during CSP with symmetry constraint to generate $\text{Crys}(\{\mathbf{v}_i\}, S, \{W_i\}, \{\mathbf{R}_i\}, \mathbf{L})$ [Figure 1(f)]. For practical implementation, we added an additional constraint ($4.0 V_a > V > 1.0 V_a$, where V_a is the volume sum of compositional atoms) to avoid the generation of unreasonable structures with extremely small or large volumes.”

Fundamentally, this work introduces 3 separate things:

1. A graph network with some new features to be the proxy model for DFT in CSP.
2. An acquisition function based on TPE in a BO for CSP.
3. An acquisition function set of symmetry constraints.

The experiments should establish which of these are important for the functioning of the system. Notably, (3) may (I can't tell because these constraints are not explained in the text) severely limit the applicability of the method because it may require a great deal of information about your target structure.

Response: We appreciate the reviewer's concise summary of our work, which has significantly helped in revising our manuscript. (1) and (2) are crucial points, while (3) has not been explained previously. In fact, inspired by this comment, we have reorganized our code and the manuscript and summarized our approach as a flexible framework of 'a database + a GN model + an optimization algorithm'.

During revision, the OQMD and MatB databases have been separately used to train the GN model and random searching (RAS), particle-swarm optimization (PSO) and Bayesian optimization (BO) have been implemented as OAs. The performance of the different combinations has been investigated and compared. Symmetry constraint is enforced during structural generation. The results show a database, GN model, optimization algorithms, and symmetry constraint for the accuracy and efficiency of GN-OA approach for CSP.

Regarding the reviewer's comment "*Notably, (3) may (I can't tell because these constraints are not explained in the text) severely limit the applicability of the method because it may require a great deal of information about your target structure.*", as explicitly explained in 'Results/Symmetry constraint' and Figure 1(e), we do not need any prior information regarding the target structure. Space group and Wyckoff position are taken as the optimizing variables in optimization algorithms during structural searching.

I have a significant number of smaller comments on the manuscript, but given the large issues above, I will wait until a potential follow on review before going into those.

Response: We thank the reviewer for the insightful comments and are eager to further exchange ideas.

Lastly, I have two comments for the authors that I would not necessarily consider part of refining this work, but I think may be useful in the future.

1. There is a body of work in using ML proxies for DFL calculations in the force field / molecular dynamics literature. There are useful ideas, especially around when to run the DFT calculations if the model is uncertain. This paper is a good example of this:

Smith, Justin S., Benjamin Nebgen, Nithin Mathew, Jie Chen, Nicholas Lubbers, Leonid Burakovsky, Sergei Tretiak, et al. 2021. "Automated Discovery of a Robust Interatomic Potential for Aluminum." *Nature Communications* 12 (1): 1257.

Response: We appreciate the reviewer for providing information regarding the ML potential and its possible combination with the GN-OA approach. We have added the corresponding discussion in the section “The limitations and future work.”

“Database. In this study, we adopted and compared OQMD and MatB databases, mainly containing stable or metastable structures (global or local minimums in PES). However, during the structural searching process, most structures are unstable (away from the minimums). The addition of the energetic data of these unstable structures should help the model to capture the entire PES landscape, thereby improving the efficiency and accuracy of GN-OA approach. Notably, generating energy landscape on numerous unstable structures is a necessary step for generating ML potential⁴². In principle, CSP based on ML potential should be more accurate; however, ML potential is generated on fixed types of elemental combinations [$\{v_i\}$ is constant], for example of aluminum⁴², while the GN model for CSP should be universal for all elements [$\{v_i\}$ is a variable]. For constant $\{v_i\}$, 6000–10000 DFT calculations are required to generate an applicable ML potential. It is an open question that how many DFT data are required to generate a reasonable GN model and how to combine existing DFT data trained for ML potential generation into GN model. This requires further investigation.”

2. I note that outside of the -SYM cases, symmetries and invariances may not be given to the model in any way. For example, rotations of the unit cell will look like different inputs to the model. Often, making these symmetries/ invariances structurally available to the model (meaning that it doesn't have to learn them from data) or using data augmentation would be worth considering in the future.

Response: Symmetries and invariances are included in the current GN model. The bond features are established on the connectivity between two atoms. Cell rotation or symmetry permutation may impact the absolute values of atomic coordinates but do not change the features, and thus, the GN model.

Reviewer #3 (Remarks to the Author):

In this paper, the authors address the problem of crystal structure prediction. Rather than using DFT, they train a graph neural network surrogate energy function on data from the Open Quantum Materials Database. Being able to solve crystal structures without DFT would be a major accomplishment, but I am not convinced that the paper accomplishes this in a convincing way, on a sufficiently hard problem.

As such I think the paper is fit for publication in its current form. In particular I have concerns about the preparation of the training set, the analysis provided in the paper and the fact that statements about computational costs seem exaggerated.

Response: We thank the reviewer for the assessment of our work, and we have addressed the reviewer's concerns below.

My main issue with the paper is that there is only very poor comparison with existing methodology. I have little personal experience of crystal structure determination with ab initio random structure search or similar methods, but my understanding is that it is often easy to find structures with small number of optimizations, for example consider fig. 7 from the AIRSS paper (<https://arxiv.org/pdf/1101.3987.pdf>). This figure is showing that for unit cells with less than 20 atoms, it is often possible to find a structure with a handful of DFT optimizations. I would like the authors to compare their method with existing methods from the literature, clearly stating what the mean number of DFT evaluation required to find the minimum are for each method.

Response: We thank the reviewer for raising this crucial issue. Figure 7 in the AIRSS paper (<https://arxiv.org/pdf/1101.3987.pdf>) shows the results from the random structure searching (RSS) [AIRSS without “AI” (ab initio)] based on the empirical Lennard-Jones (LJ) potentials, rather than the Hellman-Feynmann force derived from the DFT calculation. The oversimplified LJ potential may significantly reduce the number of attempts.

For a unit cell containing around 10 atoms, DFT-based structural searching usually requires 10^2 optimizations to find the GS. For example, 70 DFT structural relaxations are required to find the zinc-blend structure of GaAs (8 atoms in the cell) [Phys. Rev. B 75, 104113 (2007)]. Further, 120 DFT structural relaxations are required to find the α -quartz structure of SiO₂ (6 atoms in the cell) [Phys. Rev. B 82, 094116 (2010)]. We also show the process of DFT-PSO approaches to search the crystal structures of 8-atom Si and 10-atom CsPbI₃, as shown in Figure S9. They require 60-80 DFT calculations to find the GSS.

Regarding the reviewer’s comment “*My main issue with the paper is that there is only very poor comparison with existing methodology*”. We have added a section “In comparison to DFT-based approach.” in results section as follows:

“In comparison to DFT-based approach. Accuracy and efficiency are two criteria for a CSP approach. It should be noted that the accuracy of the current GN-OA approach is inferior to that of the DFT-based approach in terms of non-100% prediction accuracy and the variation of lattice parameters. In fact, the GN model is trained based on the DFT-calculated data; thus, it cannot surpass the accuracy of DFT results. In compromise with the accuracy, GN(MatB)-BO finished those tasks with much higher efficiency than DFT-based CSP, as shown in Figure 5. Here, we compare the computational cost of DFT-PSO and GN(MatB)-BO to predict 25 compounds and found that GN(MatB)-BO has a computational cost three orders of magnitude lower than DFT-based approach. DFT-PSO typically requires 60–80 DFT calculations (Si and CsPbI₃ as the example shown in Figure S9) to find the GSS, which is consistent with previous reports of 70 and 120 DFT structural optimizations to find the GSS of GaAs (8 atoms in the cell)¹⁰ and SiO₂ (6 atoms in the cell)¹², respectively.”

It was noted that during the revision process, we noted a similar work of ‘generating the periodic structure of stable materials’ using an ML approach of Crystal Diffusion Variation Autoencoder (CDVAE) submitted to arXiv on Oct. 12, 2021 (arXiv:2110.06197v1(2021),

which was first authored by the developer of CGCNN. We addressed a similar problem by using a different framework of ML approaches.

I also have some specific questions that need addressing:

1. Did the 24 binary complexes were not included in the training set? The text is ambiguous, could the authors make it crystal clear that models were neither trained nor selected (validated) based on their performance on the 24 binary complexes or on the carbon, silicon or CsPbI3 discussed in the supplemental information. Using such data in the training set would pollute the conclusions.

Response: We did not include 24 binary compounds (29 in revised manuscript) in the training. In the revised manuscript, we clearly state this in the section of “Results/Database and data split”:

“In all the training, validation and test process, the data of 29 binary compounds studied in this work, have been excluded.”

2. Fig S2. vs Fig 2. The errors on the RAS are still much larger than on the training set. This seems to suggest that the GN are very overfit on non-equilibrium structures. Does this not detract from our trust in the models? Would it be possible to show the results in S2 in the same way as Fig 2 to show how the model performs far from the training set.

Response: Figure S2 is re-plotted in the same manner as Fig 2 and shown as Figure R3 (a-b) in below. The reviewer is correct that the performance of the GN model is inferior for non-equilibrium structures, in particular, for structures with higher energy. This is because the databases (OQMD or MatB) used contained all equilibrium stable/metastable structures. Notably, the crystal structures in Figure R3(a-b) were generated without a symmetry constraint. The corresponding results are shown in Figure R3(c-d) with the crystal structures generated with a symmetry constraint. Similar trends were observed, and the MAE in the same energy scale for SYM was lower than that for noSYM.

Figure R3. GN-predicted and DFT-calculated formation enthalpies for random CaS structure (a, b) without and (c,d) with the symmetry constraint. (a)(b) and (c)(d) are shown at different energy scales.

We have added a discussion on the limitation and future improvement in the section “Limitations and future work” as follows.

“Database. In this study, we adopted and compared OQMD and MatB databases, mainly containing stable or metastable structures (global or local minimums in PES). However, during the structural searching process, most structures are unstable (away from the minimums). The addition of the energetic data of these unstable structures should help the model to capture the entire PES landscape, thereby improving the efficiency and accuracy of GN-OA approach. Notably, generating energy landscape on numerous unstable structures is a necessary step for generating ML potential⁴². In principle, CSP based on ML potential should be more accurate; however, ML potential is generated on fixed types of elemental combinations [$\{v_i\}$ is constant], for example of aluminum⁴², while the GN model for CSP should be universal for all elements [$\{v_i\}$ is a variable]. For constant $\{v_i\}$, 6000–10000 DFT calculations are required to generate an applicable ML potential. It is an open question that how many DFT data are required to generate a reasonable GN model and how to combine existing DFT data trained for ML potential generation into GN model. This requires further investigation.”

3. Fig 3: why does GN-BO-SYM immediately find the minimum? It looks to me in figures S5-S10 that GN-BO and GN-BO-SYM often (always) produce different structures and that the structures that include symmetry information always look more sensible.

Response: GN-BO did find the ground state structure, although it looks different from the structure that was found using GN-BO-SYM. This difference is a visual discrepancy due to the *periodic invariance*, i.e., there are infinite different ways of choosing unit cells with different shapes and sizes, as shown in the schematic Figure R4 below. Since GN-BO has no symmetry constraint, the cell shape can vary, and it seems significantly different from the conventional cell derived from GN-BO-SYM.

Figure R4: The same two-dimensional square lattice can have different unit cell shapes as shown by $(\mathbf{a}_1, \mathbf{b}_1)$, $(\mathbf{a}_2, \mathbf{b}_2)$, $(\mathbf{a}_3, \mathbf{b}_3)$, $(\mathbf{a}_4, \mathbf{b}_4)$, and $(\mathbf{a}_5, \mathbf{b}_5)$.

Considering the fact that most crystal structures have symmetry operations, in the revision,

the symmetry constraint is enforced. Conventional cell (each space group has its typical conventional cell) was used with the symmetry constraint in the revised manuscript. Thus, the problem above was automatically solved.

Could symmetry information also be used in the random search or particle swarm approaches? In fact, how exactly is the symmetry method implemented. I am guessing the authors are using information about the point group of the ground truth structure, but this should be stated explicitly.

Response: Symmetry constraint can also be used in random search and particle swarm approaches, as demonstrated in the revised manuscript. However, their performances are inferior to Bayesian optimization, as shown in Figure 3.

We have added a section titled “Results/Symmetry constraint” in the revised manuscript to explicitly state the how symmetry constraint has been applied. Basically, we do not require prior information about the target structure (For a good CSP algorithm, any information about the ground truth structure should not be given as a prior) but extensively searched all 229 space groups (excluding P1). In revised Figure 1(e), we have also added an image to show how symmetry constraint is applied.

“Symmetry constraint. The wealth of experimental data shows that most of crystal structures at low temperature have symmetry operations³² and adding symmetry constraint would accelerate CSP. Meanwhile, most crystal structures in training data, either OQMD or MatB, are symmetrical (with space group spanning from P2 to P230). In this work, we treat CSP with symmetry constraint, by adding two additional structural features, crystal symmetry S and the occupancy of Wyckoff position W_i for the i th atom, which are chosen through 229 space groups and associated 1506 Wyckoff positions³³. As shown in Figure 1(e), the procedure firstly chose a symmetry S among P2 to P230 and then the lattice parameters \mathbf{L} are generated within the chosen symmetry. Secondly, a combination of Wyckoff positions $\{W_i\}$ at given symmetry is selected to meet the number of atoms given in a cell. The atomic coordinates $\{\mathbf{R}_i\}$ are then given by the selected Wyckoff position $\{W_i\}$ and lattice parameters \mathbf{L} . Space group S and corresponding $\{W_i\}$ are variables upon optimizations during CSP with symmetry constraint to generate $\text{Crys}(\{\mathbf{v}_i\}, S, \{W_i\}, \{\mathbf{R}_i\}, \mathbf{L})$ [Figure 1(f)]. For practical implementation, we added an additional constraint ($4.0 V_a > V > 1.0 V_a$, where V_a is the volume sum of compositional atoms) to avoid the generation of unreasonable structures with extremely small or large volumes.”

4. I find it difficult, from the materials included in the paper, to determine how close the structures predicted by the different methods differ from the DFT ones for the different models. From Fig 3, it looks like only GN-BO-SYM is able to find the same minimum as a DFT based method. From Figs S5-S10 I am not sure what the DFT structures would be.

Response: In the revision, we reorganized Figure 3 and Figures S1-S8 (corresponding to S5-S10 in the original version). Since the symmetry constraint is enforced, the structures can

easily be compared in the revised version. Moreover, a new Figure 4 has been added to quantitatively show the discrepancies of the lattice constants and formation energies predicted by the GN(MatB)-BO approach compared with the DFT-calculated results.

Also the shifting of the energy scales makes it difficult to assess how different those optimized structures really are in terms of the energy evaluation from the GN. Ideally it would be good to know what the ΔH values that each optimization method reaches according to the graph net model and what differences with respect to the ‘true’ DFT structure this translates into when evaluated with DFT itself.

Response: In the revision, we showed the absolute energy instead of the shifting energy in the Figures. A new Figure 4 has been added to quantitatively show the discrepancies of the lattice constants and formation energies predicted by the GN(MatB)-BO approach compared with the DFT-calculated results.

5. Does Figure 4 contain the cost of a single DFT optimization to be performed after each GN-BO run?

Response: No. In the revision, when symmetry is enforced, the last-step DFT optimization is not required, and the computational cost is purely from GN-OA as shown in revised Figure 5.

The main text states that ‘We further used DFT calculations to relax the GN-BO predicted structures and obtain the precise crystal structures.’. In fact the statement that the computational cost is three orders of magnitude lower than for a DFT based method seems an exaggeration.

Response: We agree with the reviewer that this statement was an exaggeration in the previous manuscript. In the revised manuscript, as the last-step DFT relaxation is waived, it is safe to claim the computational cost to be three orders of magnitude lower than that for a DFT-based method, as shown in revised Figure 5.

For example, in the case of CaS the authors show simply running DFT on 18 canonical starting structures, leads to the correct structure being determined three times out of 18. This means that one would expect for the cost to be approximately 3 DFT optimizations to find this crystal structure. On the other hand the GN-BOSS method performs at least a single DFT optimization. Could the authors clarify this point?

Response: Since the last-step DFT optimization is waived in the revised manuscript, the two concerns above have been resolved.

There are smaller issues to do with clarity and the possibility to replicate the authors work, but these are not important unless the authors can show that the points above can be resolved.

Response: We are glad to have further exchange ideas with the reviewer.

REVIEWER COMMENTS

Reviewer #1 (Remarks to the Author):

I am satisfied with the changes made by the author. I feel the manuscript is now ready for publication.

Reviewer #2 (Remarks to the Author):

The revised manuscript is greatly improved, with cleaner experiments and a much clearer exposition. I now think this is now very close to ready for publication.

I have two scientific points on the experiments I would like addressed and some minor points on presentation

1)

For the experiment results reported in Table 1, there are two distinct reasons that the correct structure was not found.

First is a failure of the GN model to put the correct structure at the lowest energy. This means that even if the correct structure was visited during the optimization procedure, a different (wrong) structure would be selected as lowest energy.

Second is a failure of the optimization procedure. That is, the GN model ranks the correct structure lowest energy, but the optimization procedure did not visit this structure.

The two failure modes have different remedies. You can provide *some* evidence which one it is simply by comparing the GN estimated energy of the structure found during optimization to the GN estimated energy of the correct structure. I say *some* evidence because there is some ambiguity. If the GN estimated energy of the correct structure is lower, there is a failure of optimization procedure (second type). If the GN estimated energy of the correct structure is higher, there is a failure of the GN (first type). In both cases, there may be problems with the other component as well.

This should be a simple analysis that would greatly add to the value of the paper.

2)

The results are only presented as the number of cycles needed to find the target structure. But this is not quite a fair comparison because the BO method is the only one that trains a proxy model in

the inner loop. In the other words, the BO method will be much more costly per step than RAS or PSO. This should at least be acknowledged by showing the amount of computation time per step or setting a fixed computation budget and letting each method take as many steps as it can in that amount of compute. If the cost of the BO model is trivial compared to the cost of the model inference, then that claim should be made. Brief technical aside: the cost per step of the BO procedure will grow as the optimization continues because the training data grows and GPs usually do not have very favorable scaling.

3)

A few minor comments

* line 85: I am worried that you are implying something you don't intend. Black box is sometimes used in the optimization literature to mean that gradients are not available. In fact, you have gradients available to you, but you are not using them Overall, avoid black box in this scenario

* line 112: You use the word "bond" which is not well defined in a crystal system unless you define it in some way. I think you are just doing all atom pairs anyways. If so, just say that and don't call it a bond.

* line 114: I don't know what the symbols $N_{\{v\}}$ and $N_{\{e\}}$ mean and they are not defined

* Figure 1 and b: While the tag cloud to show the relative prominence of elements is cute, it's not a quantitative visualization and either supplemental data needs to provide the full details or you can find a more quantitative way to show this.

* Section starting on line 139: I think it would be worth being explicit about the invariance of the model. I think the MEGNet based model means that the output is invariant to the order of the atoms and the frame of reference that the coordinates are in, but please confirm.

* Figure 2: I think the MAE is in units of meV/atom, not meV. If so, please correct the figure.

* line 207: You can choose to ignore this comment. This suggests another procedure that could be compared to BO, RAS, PSO: trying each of the prototype structures as the optimization procedure. That would only be 300 (or maybe 600 if you have to try both permutations of the atoms) queries to the GN to do (assuming I understand the prototypes)

* Figure 5: If you are going to provide computation time, you need to be explicit about the hardware that you ran this on and whether you are counting wall clock time or some other measure of CPU time.

REVIEWER COMMENTS

Reviewer #2 (Remarks to the Author):

The revised manuscript is greatly improved, with cleaner experiments and a much clearer exposition. I now think this is now very close to ready for publication.

Response: We thank reviewer for the recognition of our previous revision.

I have two scientific points on the experiments I would like addressed and some minor points on presentation

1) For the experiment results reported in Table 1, there are two distinct reasons that the correct structure was not found. First is a failure of the GN model to put the correct structure at the lowest energy. This means that even if the correct structure was visited during the optimization procedure, a different (wrong) structure would be selected as lowest energy. Second is a failure of the optimization procedure. That is, the GN model ranks the correct structure lowest energy, but the optimization procedure did not visit this structure. The two failure modes have different remedies. You can provide *some* evidence which one it is simply by comparing the GN estimated energy of the structure found during optimization to the GN estimated energy of the correct structure. I say *some* evidence because there is some ambiguity. If the GN estimated energy of the correct structure is lower, there is a failure of optimization procedure (second type). If the GN estimated energy of the correct structure is higher, there is a failure of the GN (first type). In both cases, there may be problems with the other component as well. This should be a simple analysis that would greatly add to the value of the paper.

Response: We thank reviewer for this insightful comment. Indeed, two types of failure modes can be found in our examples. CdS and CaS are the cases of the first and second type, respectively. CdS has four-coordination zinc-blend structure as its correct ground-state structure (GSS) while all approaches of GN(MatB)-RAS, GN(MatB)-PSO, GN(MatB)-BO find wrong GSS of six-coordination rock-salt structure, as shown in Figure R1. It is noted that the zinc-blend structure has been visited as shown in Figure R1(e), however, its formation energy (ΔH) is 0.02 eV/atom above rock-salt structure. For CaS as shown in Figure R2, the GN(MatB)-BO found the correct GSS with ΔH of -2.47 eV/atom, while the GN(MatB)-PSO find a wrong GSS with ΔH 0.06 eV/atom above, indicating a failure of the optimization procedure.

CdS

Figure R1. (a) Process of GN(MatB)-RAS, GN(MatB)-PSO, and GN(MatB)-BO to search the GSS of CdS. The corresponding structures have been shown in (b-e).

CaS

Figure R2. (a) Process of GN(MatB)-RAS, GN(MatB)-PSO, and GN(MatB)-BO to search the GSS of CaS. The corresponding structure have been shown in (b-d).

The two failure modes have different remedies. A more reliable database and a better crystal representation are required to train an accurate GN model that can put the correct structure at the lowest energy. An compatible optimization algorithm to the trained GN model is required to rank the correct structure as the correct energy level. Those discussions are consistent with the discussions in the section of ‘*Limitations and future work*’.

In revision, we added a clear statement on two failure modes in the section of ‘*Limitations and future work*’, as also copied below, to convey a clearer picture of our data. The Figure R1 and R2 have been added as Figure S12 and S13, respectively, in Supporting Information. We thank reviewer for this nice idea.

“These three parts are all fast-developing research frontiers and certainly not perfect at present; therefore, the limitations of the current GN-OA approach are also apparent, such as the failure to predict the GSS of some crystals and the deviation of predicted lattice parameters. There are two failure modes. One is the failure of GN model to put the GSS as the lowest ΔH , such as CdS (Fig. S12), and the other is the failure of OA not visit the GSS with the lowest ΔH , such as GN-PSO for CaS (Fig. S13).”

2)

The results are only presented as the number of cycles needed to find the target structure. But this is not quite a fair comparison because the BO method is the only one that trains a proxy model in the inner loop. In the other words, the BO method will be much more costly per step than RAS or PSO. This should at least be acknowledged by showing the amount of computation time per step or setting a fixed computation budget and letting each method take as many steps as it can in that amount of compute. If the cost of the BO model is trivial compared to the cost of the model inference, then that claim should be made. Brief technical aside: the cost per step of the BO procedure will grow as the optimization continues because the training data grows and GPs usually do not have very favorable scaling.

Response: Reviewer is correct that the cost per step of BO would increase fast as the optimization continues. In this sense, we agree that direct comparison of number of steps to obtain GSS is unfair. Notably, this work intended to show GN-based CSP is much faster (about three orders of magnitude) than DFT-based approach, instead of comparing computational cost of different GN-based approaches. Therefore, we avoid to compare the efficiency but only focus on the accuracy of three GN-OA approaches. For data clarity, we revised Figure 3 to add a time-cost line to show the computational cost on the optimization steps.

3)

A few minor comments

* line 85: I am worried that you are implying something you don't intend. Black box is sometimes used in the optimization literature to mean that gradients are not available. In fact, you have gradients available to you, but you are not using them Overall, avoid black box in this scenario

Response: The word "black-box" has been deleted.

* line 112: You use the word "bond" which is not well defined in a crystal system unless you define it in some way. I think you are just doing all atom pairs anyways. If so, just say that and don't call it a bond.

Response: We changed "bond" to "pairs".

* line 114: I don't know what the symbols $N_{\{v^-\}}$ and $N_{\{e^-\}}$ mean and they are not defined

Response: $N_{\{v^-\}}$ and $N_{\{e^-\}}$ should be $N_{\{v\}}$ and $N_{\{e\}}$. A blank is added by mistake between the word " $N_{\{e\}}$ -" and "dimensional". We have correct it in revision. Sorry for this typo.

* Figure 1 and b: While the tag cloud to show the relative prominence of elements is cute, it's not a quantitative visualization and either supplemental data needs to provide the full details

or you can find a more quantitative way to show this.

Response: The quantitative number of elemental appearance for two databases have shown below and also in Figure S1 and S2, respectively, in Supporting Information.

Figure R3. The number of elemental appearance in MatB.

Figure R4. The number of elemental appearance in OQMD.

* Section starting on line 139: I think it would be worth being explicit about the invariance of the model. I think the MEGNet based model means that the output is invariant to the order of the atoms and the frame of reference that the coordinates are in, but please confirm.

Response: Reviewer is correct. We have added a clear statement of this invariance in the section of “GN model” as copied below.

“Since the symmetries and invariances are included in the current GN model and the pair features are established on the connectivity between two atoms. Cell rotation or symmetry

permutation would not change the features, and thus, the GN model.”

* Figure 2: I think the MAE is in units of meV/atom, not meV. If so, please correct the figure.

Response: Thank you for pointing out this. The unit in Figure 2 has been corrected.

* line 207: You can choose to ignore this comment. This suggests another procedure that could be compared to BO, RAS, PSO: trying each of the prototype structures as the optimization procedure. That would only be 300 (or maybe 600 if you have to try both permutations of the atoms) queries to the GN to do (assuming I understand the prototypes)

Response: This is an interesting comment. Such approach may be efficient but confine the GSS within prototype structures at given database, therefore, not able to predict new possible structure. Although the examples shown in current work are one of prototype structure, our objective is to predict new structures in future.

* Figure 5: If you are going to provide computation time, you need to be explicit about the hardware that you ran this on and whether you are counting wall clock time or some other measure of CPU time.

Response: We provided CPU info [Intel(R) Xeon(R) Silver 4210 CPU @ 2.20GHz] in revised Figure 5. It is wall clock time.

REVIEWERS' COMMENTS

Reviewer #2 (Remarks to the Author):

Thank you to the authors for the clear responses and iteration.

I believe the manuscript is ready for publication.